# Spatial Distribution and Sustainable Development of Living Woody and Coarse Woody Debris in Warm-Temperate Deciduous Broadleaved Secondary Forests in China

**DOI:** 10.3390/plants13050638

**Published:** 2024-02-26

**Authors:** Fang Ma, Shunzhong Wang, Weiguo Sang, Shuang Zhang, Keming Ma

**Affiliations:** 1State Key Laboratory of Urban and Regional Ecology, Research Center for Eco-Environmental Sciences, Chinese Academy of Sciences, Beijing 100085, China; 2State Key Laboratory of Vegetation and Environmental Change, Institute of Botany, Chinese Academy of Sciences, Beijing 100093, China; 3College of Life and Environmental Sciences, Minzu University of China, #27 Zhongguancun South Avenue, Beijing 100081, China

**Keywords:** spatial distribution pattern, size class, habitat heterogeneity, warm-temperate secondary forest, sustainable development

## Abstract

The investigation into the spatial patterns of living woody (LWD) and coarse woody debris (CWD) in warm-temperate deciduous broadleaved secondary forests serves as a foundational exploration of the mechanisms governing coexistence and mortality in forest ecosystems. The complete spatial randomness null model (CSR) was employed to analyze spatial distribution patterns, with the independent component null model (IC) and canonical correspondence analysis (CCA) utilized to elucidate spatial correlations and topographic influences. All three models were applied to LWD and CWD across various size classes within a 20-hectare plot in the Dongling Mountains. The study’s findings indicate that both LWD and CWD predominantly exhibited aggregated patterns, transitioning to a random distribution as the size class increased. Both increasing abundance and maximum diameter at breast height (DBH) also have a significant influence on the distribution of species. Notably, rare species exhibited higher aggregation compared to common and abundant species. The spatial correlation results between LWD and CWD across various size classes predominantly showed positive correlations and uncorrelated patterns within the sampled plots. CCA analysis further revealed that elevation, convexity, slope, and aspect significantly influenced the spatial patterns of LWD and CWD across different size classes. Within the sample site, trees display a tendency to grow and die in clusters. Biotic factors have a more significant influence on species distribution than abiotic factors.

## 1. Introduction

The examination of species spatial distributions in forest ecosystems and the underlying mechanisms that shape them have always been a prominent subject of research [1,2,3]. The spatial distribution pattern not only serves as a quantitative representation for assessing community structure but also plays a pivotal role in reflecting the dynamic succession of species across various growth phases. This aids in elucidating the origins of coexistence mechanisms within ecological communities [4,5,6,7].

The spatial patterns of forest stands are influenced by ecological processes such as growth, reproduction, and mortality [8,9]. Coarse woody debris (CWD), referring to decomposing organic material remaining after tree death, is prevalent in various natural forest ecosystems [10]. Additionally, it serves as a crucial substrate for the process of natural regeneration [11,12] and plays a significant role in the establishment, development, and succession renewal of forest ecosystems [13]. Furthermore, coarse woody debris (CWD) creates voids or gaps within the forest ecosystem, fostering the expansion and rapid growth of light-dependent species. This, in turn, influences the overall composition and structural dynamics of the community. Numerous investigations have explored the spatial distribution of CWD and factors contributing to its mortality [14,15,16]. Nevertheless, the aforementioned studies primarily concentrated on dead trees alone, with limited studies comparing spatial relationships and patterns between LWD (living woody) and CWD.

The spatial distribution pattern of forests is intricately determined by a combination of species’ attributes, interspecific correlations, and environmental circumstances [17]. Multiple investigations have demonstrated that dispersal limitation, negative density dependence, and habitat heterogeneity represent three pivotal factors influencing the spatial distribution pattern of species [2,18,19,20]. The prominence of opportunistic factors in species distribution is underscored by dispersal limitation [21]. Negative density dependence suggests that species growth is regulated by negative feedback mechanisms, curbing the formation of single-optimistic communities [22]. Habitat heterogeneity indicates that environmental filtering leads to the aggregation of species in suitable habitats, with ecological niche differentiation contributing to species coexistence [18]. The existence of intricate spatial processes introduces the possibility that spatial patterns at various scales may be attributed to distinct causes [1]. Species’ attributes, spatial associations, and microtopographic factors are frequently overlooked or insufficiently addressed, particularly in the context of analyzing spatial patterns related to CWD.

The spatial distribution of forests not only displays deviations among different species but also demonstrates significant disparities depending on the growth stage [23]. Furthermore, it consistently highlights the need for analyzing the spatial distribution of trees with regard to the different phases of development in forests [24]. Plotkin et al. conducted a study that reported that the aggregation intensity has been shown to exhibit variation across different phases of growth [25]. Comita et al. [26] and Lai [27] demonstrated that the same species indicates diverse responses to environmental conditions during distinct growth stages. 

Previous research has demonstrated variances in mortality patterns among trees with varying diameters [15]. The spatial pattern of plant individuals at different developmental stages and their connectivity results from long-term interactions between plant communities and their surrounding environment [28]. Analyzing spatial patterns in conjunction with different growth stages aims not only to showcase the characteristics of spatial and temporal distribution in forest stands but also to elucidate the underlying mechanisms enabling species coexistence [29]. Nevertheless, there is a scarcity of research examining spatial patterns throughout various phases of growth, especially in conjunction with CWD. 

In recent decades, significant scholarly inquiry has been dedicated to investigating the mechanisms impacting spatial distribution and the ecological processes that underlie these patterns. Prior research has predominantly concentrated on tropical rainforests [1,28,30,31] and subtropical forests [32,33], with relatively little emphasis on warm-temperate [34] and temperate forests [9]. Moreover, it is imperative to acknowledge that different forest types exhibit unique spatial distributions. For example, habitat heterogeneity emerged as a notable factor influencing the spatial distribution of woody plants in both tropical forests (BCI sample site in Panama) and subtropical forests (Fushan sample site in Taiwan). However, no discernible impact of habitat heterogeneity on the spatial pattern of woody plants was noted in the Changbai Mountain sample site located in the northeastern temperate zone [35]. Warm-temperate forests constitute an essential category of vegetation in China, characterized by abundant plant resources and notable geographical distribution. Due to extensive anthropogenic activities, the presence of natural forests has significantly diminished. The warm-temperate deciduous broadleaved forests in Beijing primarily consist of secondary forests that have undergone a process of recovery following significant damage. The dataset obtained from the 20–hectare warm-temperate dynamic monitoring sample plot located in the Dongling Mountains (DLM) offers a valuable opportunity to investigate the dominance pattern of warm-temperate forest species and the mechanisms underlying the maintenance of species assemblages. 

The primary aims of this study were to (1) analyze and interpret the spatial distribution patterns of LWD and CWD across different size classes in a 20 hm^2^ warm-temperate secondary forest; (2) investigate the role of biotic and abiotic factors in the spatial distribution patterns of LWD and CWD; and (3) compare the reasons for differences and similarities in spatial patterns of LWD and CWD. Our results are expected to provide helpful insights into the dynamic patterns of warm-temperate secondary forests. It intends to enhance the managerial capabilities of forest managers and establish the fundamental framework for the sustainable use and development of local regional forests.

## 2. Materials and Methods

### 2.1. Study Site

The study was conducted at the Beijing Xiaolongmen Forest Park Reserve (39°48′34″–40°10′37″ N, 115°25′–116°10′07″ E). The vegetation here is a typical warm-temperate and deciduous broadleaved secondary forest, with relatively complex community structures. The dominant tree species in the main forest layer, known as the tree layer, include *Acer mono*, *Quercus wutaishanica*, *Populus davidiana*, *Betula platyphylla*, among others. The sample location exclusively features the North China larch (*Larix principis*–*rupprechtii*) as the sole coniferous species, commonly found in the north temperate zone. *Abelia biflora*, a species in the understory, exhibits a unique distribution spanning East Asia and Mexico. Within this region, *Syringa pekinensis* and *Syringa pubescens* exemplify vegetation with a temperate distribution typical of the Old World. Furthermore, the combination of plant species from both northern and southern regions results in the presence of tropical and pantropical distributions, indicating that the sample location is situated in a transitional zone.

The Donglingshan Mountain sample plot has a warm-temperate continental monsoon climate with four distinct seasons, with a mean annual temperature of 4.8 °C, an average July (hottest) temperature of 18.3 °C, and an average January (coldest) temperature of −10.1 °C. The annual frost-free period is about 195 days, and the annual sunshine is about 2600 h. Annual precipitation in the study area ranges between 500 and 650 mm, with June and August accounting for roughly 78% of total precipitation. Mountain brown soil is the parent soil material [36].

A 20 ha (400 m × 500 m) plot (40°00′ N, 115°26′ E) was established in Xiaolongmen Forest Park Reserve in 2010, with the first census completed in 2010 following the standard field protocol of the CTFS (Center for Tropic Forest Sciences, Condit 1995, http://www.ctfs.si.edu, accessed on 30 November 2023). The plot is characterized by rugged terrain (Figure 1): the elevation varies from 1298.21 m to 1506.34 m and the slope ranges from 8.46° to 48.49°, with a mean of 31.98°. 

### 2.2. Data Collection

The plot (20 ha/400 × 500 m) was divided into 500 subplots measuring 20 × 20 m each, and all free standing trees (DBH ≥ 1) in these grids were identified, tagged, and mapped following standard field procedures [37]. A comprehensive examination of living woody debris was conducted on a total of 500 plots, resulting in the documentation of 56 species, 36 genera, and 20 families. According to the dataset from the living woody debris, the coarse woody debris was identified, tagged, and mapped with a DBH greater than or equal to 5 cm. A total of 32 species, 25 genera, and 15 families were documented (unidentified species as “unknown”). *Quercus wutaishanica* and *Acer mono* Maxim. are absolute dominant species in our plot. To obtain a sufficient sample size for point pattern analyses, we chose 46 common species with no fewer than 20 individuals [38].

### 2.3. Data Analyses

The spatial point pattern has been widely used to analyze the spatial distribution pattern of species [39]. In this paper, we used g(r) to study the spatial distribution of species at 0–50 m scale and mean g_0–10_ as a measure of mean conspecific aggregation density within 10 m of a tree [28]. The g function is derived from Ripley’s K function [40], and g function is a probability density function which can effectively eliminate the cumulative effect with increasing scale in Ripley’s K function. The g(r) function replaces circles with rings, irrespective of varying scale patterns, thus minimizing the influence of large-scale patterns on smaller scales. Consequently, the g(r) function can demonstrate greater sensitivity in assessing the degree to which the observed distribution of points on a specific scale differs from the anticipated value. The g(r) function is utilized to analyze patterns on a specific scale determined by the radius r. It operates as a distance-dependent correlation function, examining the pattern across the distances of all localized pairs of individuals. The formula for its calculation is as follows:$$g(r)=\frac{1}{2\pi r}\frac{dK(r)}{d(r)}$$

The g(r) function has emerged as a crucial analytical technique for assessing spatial patterns and the degree of clustering, owing to its enhanced intuitiveness and accuracy of results. An aggregated distribution is indicated when g(r) is greater than 1, a regular distribution is observed when g(r) is less than 1, and a random distribution is identified when g(r) is equal to 1. Specifically, when g(r) > 1, it signifies that the density of points on scale r exceeds that of the random distribution, suggesting an aggregated distribution pattern. For instance, if g(r) = 2, it indicates that the density of points on scale r is two times that of the random distribution.

We use complete spatial randomness (CSR) as a null model. The null model is widely used for univariate point patterns; it assumes no interactions between points and indicates that trees can occur at any position without the influence of biological processes. Nine hundred and ninety–nine random Monte Carlo simulations are used to test whether a species is not significant from the random distribution. If the observed value falls within the 2.5th and 97.5th percentiles, the null model cannot be rejected and the species is aggregated. These simulations are performed in the “spatstat” package in R. 

We used the mean aggregation intensity g_0–10_ [28] to compare the distribution patterns of LWD and CWD across various size stages and applied regression analysis to investigate the association between abundance, maximum diameter at breast height (max DBH), and g_0–10_. The observed species abundance served as the basis for this analysis, and we divided the abundance into three levels: abundant (with abundance ≥ 1000 individuals/20 ha), common (100–999 individuals/20 ha), and rare (<100 individuals/20 ha). The data were subjected to statistical analysis to determine their significance, employing either the Kruskal–Wallis or Wilcoxon rank-sum test. According to the empirical assessment of diameter at breast height (1.3 m above the ground, DBH) and the identified study requirements, we divided the sizes into three levels: small size (<10 cm, S), medium size (≥10 and <20 cm, M), and large size (≥20, L).

The present study employed a null model of the independence of components (IC) model to examine the connections between various growth phases prior to and following tree mortality [1]. The null model of the independence of components assesses the degree to which the observed distribution of type 2 in relation to type 1 differs from the expected value of type 2. Throughout the process, the positions of type 1 points remain constant, while the collective type 2 points undergo a transformation across the research sample through the utilization of a random vector. Monte Carlo simulations consisting of 999 iterations were conducted to generate an envelope line with a confidence level of 95% and to test the statistical significance of the observed point pattern results.

The microtopographic variables used in the canonical correspondence analysis (CCA) were the elevation, slope, aspect, and convexity of each 20 × 20 m grid. In this study, we examine the correlation between microtopography and the spatial distribution of woody vegetation. The Monte Carlo permutation test was performed to evaluate the significance of these relationships using the “vegan” package in R. Each microtopographic variable was tested at the 5% significance level using 1000 random permutations. All analyses were performed using R4.3.1 (R Development Core Team) and Microsoft Excel 2023.

## 3. Results

### 3.1. Spatial Distribution Pattern of LWD and CWD in Each Size Class

The results under the CSR null model showed that aggregated distribution was the dominant pattern in the DLM plot, and the pattern varied with the size classes of LWD and CWD at 0–50 m (Figure 2). The small size of LWD did not exhibit convergence to random within 0–50 m. The medium size transitioned to a random distribution at 40 m. The large size displayed a pattern of initially randomizing and then aggregating distribution constrained within a distance of 5 m. The occurrence of randomness at 5 m could be attributed to interactions between intraspecific or interspecific individuals. The distribution pattern of CWD across various size classes likewise demonstrates a shift from aggregated to random distribution as the scale increases (Figure 2). The clustering pattern is predominant for the small and medium sizes, whereas the large size has a tendency towards random distribution at 41 m. 

### 3.2. Aggregation Intensity g_0–10_ and Attributes of LWD and CWD in Each Size Class

The investigation documented a significant reduction (*p* < 0.001) in the aggregation intensity g_0–10_ of LWD concomitant with the abundance increasing (Figure 3). Similarly, a significant negative correlation (*p* < 0.01) was found between the abundance of CWD and g_0–10_ (Figure 3). Furthermore, the aggregation intensity g_0–10_ exhibited a negative linkage with the maximum DBH of both LWD and CWD. Specifically, LWD was found to be significant but CWD was non–significant (Figure 3).

The aggregation intensity g_0–10_ of LWD exhibited a decrease as the size class increased (Figure 4), and there were significant variations observed between the three size levels (Kruskal–Wallis χ^2^ = 11.51, *p* < 0.01). Conversely, the trend for CWD was statistically non-significant across various size classes (*p* > 0.05). 

The results of the Kruskal–Wallis test indicated a statistically significant variation in the three abundance levels of LWD (χ^2^ = 36.95, *p* < 0.001), and the aggregation intensity g_0–10_ of rare species was substantially greater than the common and abundant species. This finding provides additional evidence supporting the negative correlation between abundance and g_0–10_. The sample size of rare species within CWD is insufficient for meaningful comparison.

### 3.3. Spatial Correlation Analysis of LWD and CWD

The spatial interactions among trees of varying size classes displayed considerable variability (Figure 5). Within a 50 m scale, the association between the small and medium size of LWD underwent a transition from a positive correlation to no correlation at a distance of 38 m. This transition suggests an attraction between these two size classes at a smaller spatial scale, with the relationship becoming unrelated as the scale increases. At a distance of 31 m, a negative correlation was observed between small–sized and large-sized trees, indicating a shift from mutual exclusion to non-correlation.

Moreover, a positive association was seen between the small size of LWD and CWD with varying sizes. This observed positive correlation may be attributed to the favorable conditions created by CWD at a smaller spatial scale, fostering the colonization, growth, and development of saplings. The linkage between the medium and large sizes experienced a shift from a negative correlation to non-correlation at a distance of 10 m. The interaction involving the medium and large sizes in conjunction with CWD was predominantly characterized by non-correlation.

### 3.4. Microtopographic Effect on LWD and CWD

The results indicate that four microtopographic parameters significantly influence the distribution of various sizes of LWD and CWD (Table 1). Distinct size classes exhibited varied distribution patterns influenced by different microtopographic variables (Figure 6). Specifically, the combination factors accounted for 9.25% of the variance in the distribution of small–sized LWD. Furthermore, the spatial distribution of small–sized LWD demonstrated a significant positive correlation with elevation and a significant negative correlation with convexity, slope, and aspect along the first axis. On the second axis, the distribution of small–sized LWD was significantly positively correlated with elevation and convexity and significantly negatively correlated with slope and aspect (Figure 6; Table 1).

The investigation revealed that microtopographic factors exerted a more pronounced influence on LWD in comparison to CWD. Notably, convexity emerged as the most influential factor affecting the distribution of LWD across various size classes. Conversely, elevation was identified as the primary determinant influencing the distribution of CWD. Furthermore, while microtopographic factors demonstrated a significant impact, the magnitude of their influence on the spatial patterns of both LWD and CWD was comparatively modest.

## 4. Discussion

### 4.1. General Spatial Pattern

The aggregated pattern is prevalent at the given scales within the natural plot. Aggregation was also the primary distribution in the Dongling Mountains plot. The result is consistent with prior studies undertaken in various forest ecosystems, including tropical forests [28,41,42], subtropical forests [33,38], and temperate forests [9]. Within the context of natural forest ecosystems, a discernible trend emerges, indicating a significant decrease in aggregation intensity during the transition from juvenile to adult stages, subsequently manifesting as an aggregates–random–regular distribution [43]. The Dongling Mountains sample site also validates these observations, illustrating a reduction in aggregation intensity for both living woody (LWD) and coarse woody debris (CWD) as size class increases. Moreover, the distribution pattern undergoes a shift from aggregation to randomization. 

The alteration in spatial patterns within the forest stands primarily resulted from two factors: the abundance of small-sized trees and the notable clustering that emerged during the developing stage [44]. During the development phase of forest dwellers, a noteworthy mortality incidence occurs among juvenile trees, with only a minority successfully transitioning into adulthood. This transition is accompanied by a discernible reduction in aggregation processes. Conversely, subsequent to the canopy growth phase, mature trees attain a state of equilibrium, where the distribution of canopy trees is predominantly influenced by individual mortality [45]. The distribution pattern of CWD observed across different growth phases serves as an indicator of both tree mortality and disturbance dynamics within the community. In the current study, the examined stands endured an extended period of undisturbed conditions. The observed clustering of CWD across diverse size classes may be attributed to mortality influenced by density-dependent and successional processes.

The fluctuations in spatial pattern might be seen as an indication of the survival strategies or adaptive mechanism employed by a population [28]. One potential advantage of aggregated patterns lies in their ability to alleviate the adverse impacts of competitively dominant species on disadvantaged species, thereby facilitating the survival and persistence of the latter within the community [34]. Conversely, during a developmental phase in the community, where there is an elevated demand for resources, the aggregation may decline as individuals disperse to acquire essential means for survival. Subsequent to the completion of environmental screening and habitat filtering, a substantial proportion of species experiencing a competitive disadvantage undergo mortality. Consequently, the community attains a state of relative stability, aligning with the stabilization of the distribution pattern [46]. 

### 4.2. Functional Traits on Spatial Pattern of LWD and CWD

Previous studies have highlighted the substantial impact of functional traits, including abundance, size class, and dispersion limitation, on the spatial distribution of species [28,38,47]. In this study, the CSR null model was employed to explore the relationship between species’ functional traits (abundance and max DBH) and aggregation intensity (g_0–10_) at the 0~10 m scale for both LWD and CWD within the sampled plots.

The findings reveal a noteworthy negative correlation between g_0–10_ and both the abundance and max DBH of LWD. This aligns with earlier research indicating a decrease in forest aggregation indices with increasing abundance [9,28], a trend also observed in subtropical evergreen deciduous broadleaved mixed forests [33]. In our specific sample site, g_0–10_ was higher for abundant and common species at the same scale. This observation resonates with similar occurrences documented in other forest ecosystems [48]. These patterns may be attributed to elevated mortality rates resulting from intraspecific competition and density-dependent effects exhibited by abundant and common species [49], aligning with the documented phenomena in various forest environments. 

Moreover, research has demonstrated a decline in forest stand aggregation intensity g_0–10_ as the size class increases [44]; similar conclusions were revealed in this study. Naturally regenerated forest stands often exhibit clustered spatial distributions, where smaller trees tend to cluster, while larger trees display a more regular distribution pattern [24]. The aggregation of smaller individuals contributes to the manifestation of the group effect, positively impacting population renewal and biodiversity conservation. The diminishing aggregation in larger diameter classes can be attributed to competition-induced self-thinning [50].

The findings of this study unveil a notable and negative correlation between the abundance of CWD and the aggregation intensity g_0–10_. Moreover, an inverse relationship was observed between size class and g_0–10_, and upon reaching the maximum diameter, there was no longer any association with g_0–10_. The consistency in trends between CWD and LWD suggests a relationship between the two. The more pronounced trend observed in LWD could be attributed to the fact that CWD originates from LWD and is influenced by the functional characteristics of their source trees.

### 4.3. Spatial Correlation Analysis of LWD and CWD

Research has brought to light a noteworthy association between the spatial distribution of populations and size classes, exhibiting a higher correlation at smaller sizes. However, as the diameter class increases, this correlation gradually weakens [51]. The findings of this study align with the aforementioned result, as the correlation undergoes a progressive transition towards being uncorrelated with the increase in size class. Within the specified range of 0–50 m, a positive correlation was observed between small and medium sizes of LWD, indicating attraction and aggregation. Conversely, a negative correlation was identified between small and large sizes of trees, suggesting mutual repulsion between the two. There exists a hypothesis proposing that the growth of small trees is constrained by the availability of light resources.

This suggests that within a given community, small and medium-sized trees exhibit limited competitiveness for soil moisture, nutrients, light, and other resources compared to their larger counterparts. It is proposed that the evolutionary responses developed through natural selection over an extended period necessitate that small and medium-sized trees primarily provide shelter to one another. This fosters group effects and enhances the likelihood of individual survival. Building upon the observed positive association, it may be inferred that the presence of medium-sized trees has a beneficial impact on the growth of small trees [52]. However, as individual plants undergo growth and maturation, the competition for resources among individuals at various developmental stages intensifies. This heightened competition leads to an enhanced ability to withstand environmental stresses but also weakens the protective influence between plants. Consequently, there is a tendency for a negative correlation or lack of correlation between small and large trees as well as between medium and large trees.

A positive association was identified between the diameter classes of CWD and the small size of LWD. However, the link between the diameter classes of CWD and the medium and large sizes of LWD was predominantly uncorrelated. Several potential explanations for these phenomena exist. Firstly, the creation of gaps within the forest leads to a reduction in resource competition, establishing a conducive environment for smaller organisms. Additionally, as the decomposition process progresses, the detritus serves as vital nourishment for regeneration. Finally, it has been observed that the existence of CWD has a beneficial effect on the viability and establishment of new plant growth [53]. The correlation between medium size or large size and CWD exhibited non-correlation. The correlation between medium or large sizes and CWD exhibited no significant correlation. This is likely due to the completion of ecological niche differentiation at this stage, resulting in a highly stable growth period where the contribution of CWD was not deemed significant. There is a suggestion that the correlation between populations in early successional communities is primarily negative and uncorrelated. As the process of succession progresses, the negative correlation gradually diminishes, while the positive correlation increases. This observation, in turn, confirms that the sampled stand is in an intermediate stage of succession.

### 4.4. Microtopographic Variables on LWD and CWD

Habitat heterogeneity, widely acknowledged as a crucial process influencing aggregated patterns, plays a significant role in the spatial distribution pattern of woody plants at large scales [54]. The spatial distribution of species in tropical and subtropical forests at regional scales is typically influenced by habitat heterogeneity [27,35]. Global research on forest samples has consistently shown that habitat, particularly microtopographic parameters, exerts a substantial influence on the spatial distribution of tree species [55,56]. Numerous studies have demonstrated notable positive or negative associations between multiple species and variables such as slope, elevation, or aspect [57,58]. The interaction between species and their habitat may exhibit variations across distinct stages of growth, maintaining a consistent pattern from sapling to juvenile stages but undergoing a shift at maturity [27].

The findings from the CCA indicated that four key microtopographic parameters, elevation, slope, convexity, and aspect, played a major role in explaining the spatial patterns of population with various size classes at the Dongling Mountains sample site (*p* < 0.001). As an illustration, the collective influence of these characteristics accounted for 9.25% of the distribution of small-sized species with respect to LWD. Notably, convexity exhibited the most substantial contribution to the variability, while aspect had the least significant contribution. While the total impact may be modest, it remains statistically significant, hence supporting the necessity of implementing environmental filtration on a larger scale in order to enhance its efficacy [59]. The available evidence from tropical [56], neotropical [60], and subtropical forests [27] indicates a significant correlation between the distribution of the majority of tree species and abiotic environmental factors, such as elevation. This correlation provides support for the effect of environmental filtration on the aggregation of trees.

Hu et al. observed that diffusion limitation, indicated by the neighborhood index, had the most critical role in determining the distribution of small trees, while environmental variables played the most important role in defining the distribution of large trees [61]. The current study examined the influence of microtopographic elements on varying size class. Contrary to expectations, no consistent pattern was seen between microtopographic factors and diameter class. According to the study conducted by Lan et al., it was determined that elevation showed the most significant influence on species distribution patterns compared to other topographic variables such as convexity, slope, and aspect [62]. The primary factor that exhibited the most significant impact on the LWD in this research was convexity, while elevation emerged as the main variable influencing the spatial distribution of CWD. The rationale for this study may be attributed to its focus on the distribution of various diameter classes.

The predictive ability of microtopographic parameters in accounting for the distribution of CWD across various diameter classes was found to be lower compared to that of LWD. This relationship is likely due to the fact that the distribution of LWD serves as a prerequisite and foundation for the distribution of CWD.

A precise understanding of the spatial distribution of trees within a forest stand is imperative for effective stand management and the implementation of silvicultural practices in harmony with natural principles. Consequently, managerial decisions should be tailored to the fundamental characteristics of the stand at each stage of its development [63]. The overarching objectives of these interventions are to improve the structure and composition of the stand, with specific emphasis on facilitating regeneration. In the early stage of stand development, the principal process involves self-thinning among low- and mid-story trees. Crop-tree thinning can be employed to expedite this inherent process or promote increased tree diversity. In the mature tree phase, the preservation of valuable parent trees is imperative to facilitate the establishment of new trees. Additionally, the practice of sanitary felling can be implemented to address the concern of CWD that may serve as potential habitats for pests and diseases.

## 5. Conclusions

This present investigation studied the spatial pattern of warm-temperate secondary forests in the Dongling Mountains, serving as an initial step towards comprehending the structure of plant communities and uncovering the mechanisms of species coexistence. The findings indicated that the warm-temperate secondary forest had a prevalence of aggregation. It was observed that the aggregation intensity g_0–10_ of both LWD and CWD showed a decreasing trend as abundance and size class increased. The spatial association between LWD and CWD displayed predominantly positive correlations and non-correlation. Furthermore, there was a significant correlation found between the distribution of LWD and CWD across various size classes and microtopographic variables.

Our study has unveiled that trees within the 20-hectare fixed monitoring sample plots of warm-temperate deciduous broadleaved secondary forests tend to grow and die in clusters. Biotic factors exert a more pronounced influence on species distribution compared to abiotic factors, although the impact of abiotic factors remains noteworthy. While it is premature to generalize the findings from a specific forest to all forests, the study underscores the significance of fallen wood in shaping the growth of standing trees. This highlights the potential to counteract the adverse impacts of logging practices on biodiversity. Consequently, we advocate for the classification and management of trees at various growth stages, along with the promotion of seedling nurturing and the protection of mature trees. The empirical evidence presented in this study is integral for the effective management of the spatial distribution pattern of secondary forests in the warm-temperate zone, providing a crucial reference for the sustainable management of regional forests.

## Figures and Tables

**Figure 1 plants-13-00638-f001:**
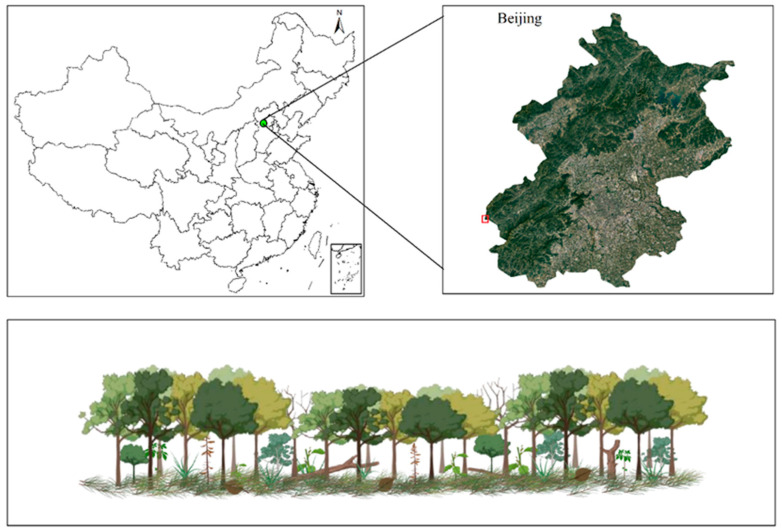
Plot location and forest stand schematic.

**Figure 2 plants-13-00638-f002:**
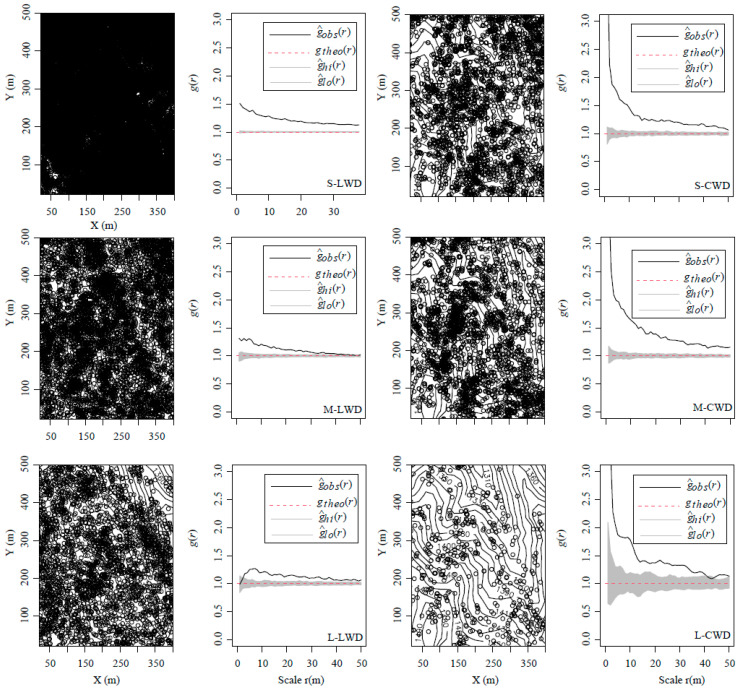
Distribution map and point pattern analysis of different development stages of LWD (live woody debris) and CWD (coarse woody debris) in the 20 ha Dongling Mountains forest plot. Left panels show the number in the contour map is elevation (m). The second panel shows the relationship between the univariate pair-correlation function (g(r)) and scale for the all species. Two rows of LWD on the left and two rows of CWD are shown on the right. The lines represent g(r); the gray areas indicate the simulation envelopes generated from 999 Monte Carlo simulations under the null hypothesis of complete spatial randomness (CSR, the middle panels). The figures were created using R 4.0.3 software (https://www.r-project.org/, accessed on 1 October 2020).

**Figure 3 plants-13-00638-f003:**
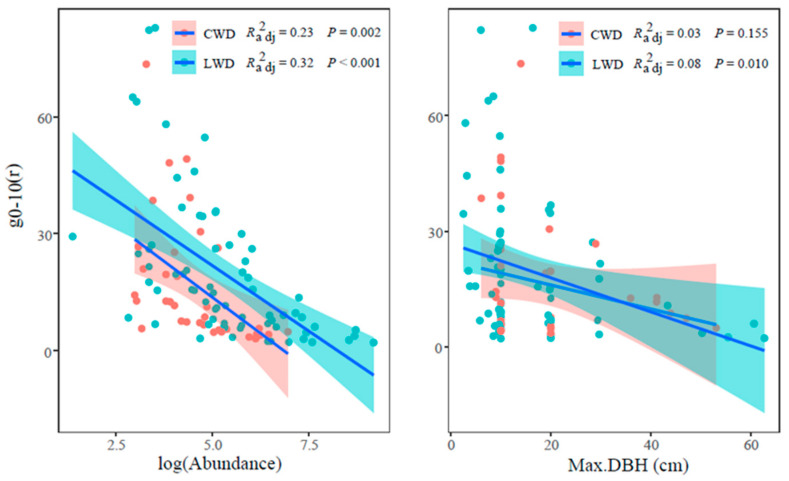
Relationship between aggregation intensity g_0–10_ and maximum DBH and abundance in the Dongling Mountains forest dynamics plot.

**Figure 4 plants-13-00638-f004:**
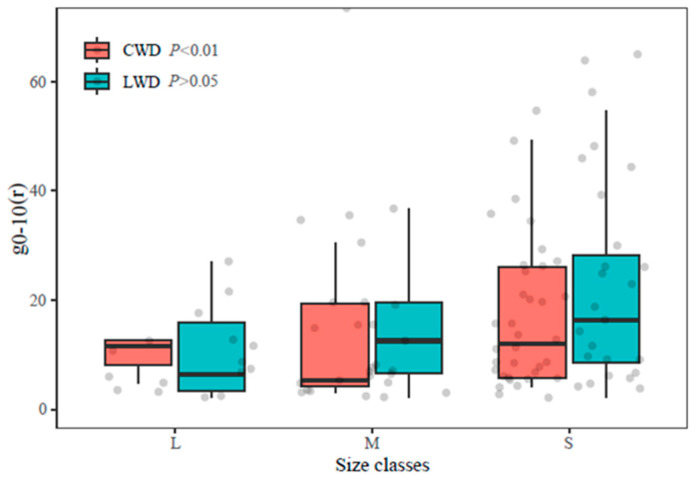
Comparison of different size classes of LWD and CWD.

**Figure 5 plants-13-00638-f005:**
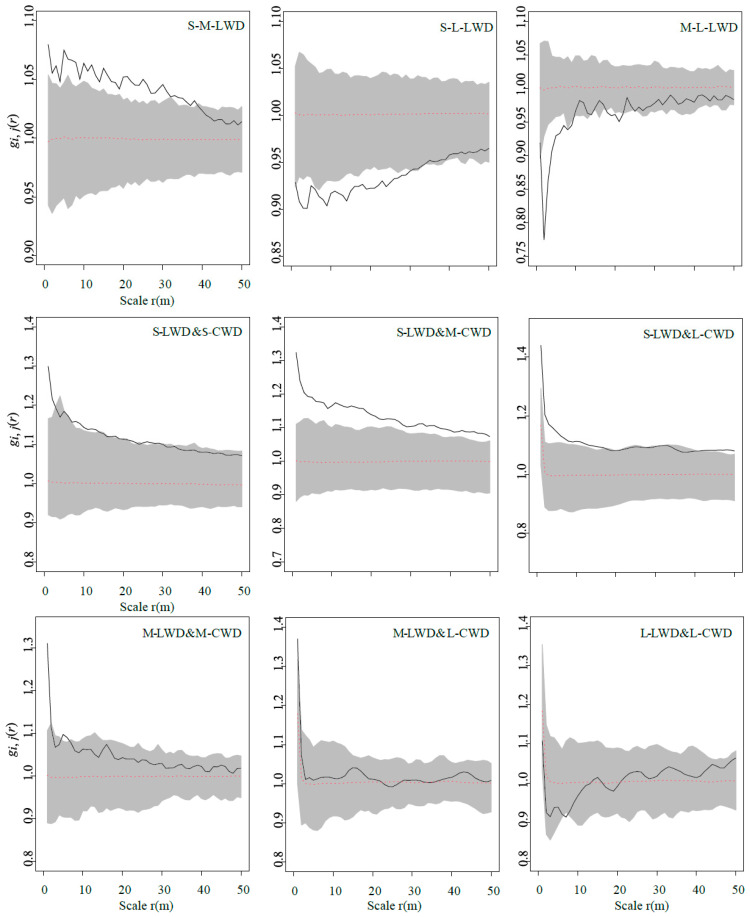
Analysis of association among different size classes of living woody and coarse woody debris. Note: S, M, L indicate different size classes.

**Figure 6 plants-13-00638-f006:**
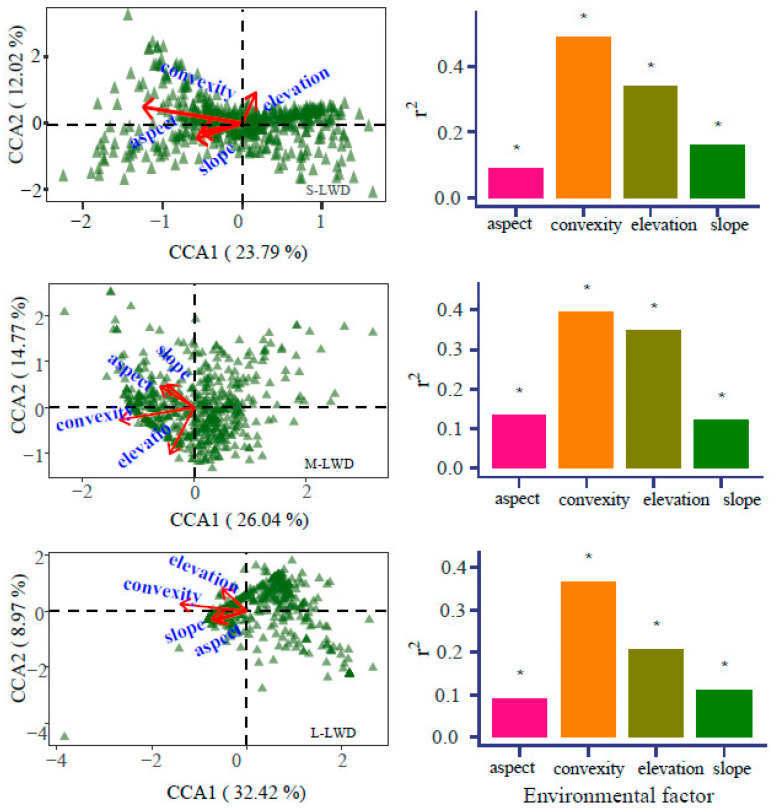
Canonical correspondence analysis (CCA) biplot of the different DBH stages of LWD and CWD species and topographic factors. Note: *p*-value < 0.05 (*).

**Table 1 plants-13-00638-t001:** Permutation test for the topographic factors explaining the distributions of LWD and CWD plants in the Dongling Mountains plot.

**LWD**	**Topographic Factor**	**CCA1**	**CCA2**	**R^2^**	**Pr (>r)**
Small size	Elevation	0.131	0.991	0.342	0.001 ***
	Convexity	−0.905	0.425	0.489	0.001 ***
	Slope	−0.735	−0.679	0.160	0.001 ***
	Aspect	−0.862	−0.508	0.087	0.001 ***
Middle size	Topographic factor	CCA1	CCA2	R^2^	Pr (>r)
	Elevation	−0.389	−0.921	0.348	0.001 ***
	Convexity	−0.973	−0.229	0.395	0.001 ***
	Slope	−0.666	0.746	0.121	0.001 ***
	Aspect	−0.778	0.628	0.135	0.001 ***
Large size	Topographic factor	CCA1	CCA2	R^2^	Pr (>r)
	Elevation	−0.698	0.717	0.206	0.001 ***
	Convexity	−0.987	0.159	0.366	0.001 ***
	Slope	−0.969	−0.248	0.112	0.001 ***
	Aspect	−0.944	−0.329	0.091	0.001 ***
**CWD**	**Topographic factor**	**CCA1**	**CCA2**	**R^2^**	**Pr (>r)**
Small size	Elevation	−0.999	0.005	0.117	0.001 ***
	Convexity	0.747	0.665	0.100	0.001 ***
	Slope	0.974	0.228	0.237	0.001 ***
	Aspect	0.882	−0.471	0.217	0.001 ***
Middle size	Topographic factor	CCA1	CCA2	R^2^	Pr (>r)
	Elevation	0.939	0.342	0.245	0.001 ***
	Convexity	0.399	0.917	0.102	0.001 ***
	Slope	−0.929	0.368	0.113	0.001 ***
	Aspect	−0.723	0.691	0.064	0.001 ***
Large size	Topographic factor	CCA1	CCA2	R^2^	Pr (>r)
	Elevation	0.956	−0.294	0.189	0.001 ***
	Convexity	0.957	0.290	0.172	0.001 ***
	Slope	−0.572	0.821	0.010	0.421
	Aspect	0.864	0.503	0.004	0.754

Note: strikingly significant difference: *p*-value < 0.001 (***).

## Data Availability

The data in this study are available from the authors upon request.

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
