# Peer review of "Spatial Distribution and Sustainable Development of Living Woody and Coarse Woody Debris in Warm-Temperate Deciduous Broadleaved Secondary Forests in China"

_plants, 2024, doi:10.3390/plants13050638_

Round 1

Reviewer 1 Report

Comments and Suggestions for Authors

The manuscript is interesting and might provide new insights into the spatial distribution of trees in forests, as well as issues related to tree competition, mortality and production.

On the other hand, I must state that the text is not well-written—some parts are difficult for readers to follow. Moreover, there are some terms that do not seem usual for forestry or forest ecology (e.g., living woody debris). Throughout the text, but especially in the Discussion section, the writing does not adhere to the principles of academic writing.

Therefore, I suggest a major review of the manuscript and the necessity to make detailed linguistic corrections by a native English speaker.

Specific comments:

The title

It is excessively long and somewhat confusing. Please reconsider it.

Abstract

The authors should specify the types of forests, including the species compositions included in the study. Additionally, a strong concluding or generalizing sentence is missing at the end of the Abstract section.

Introduction

In the first sentence, clarify whether the authors mean tree species specifically or all flora.

"Living woody debris" sounds unusual for traditional scientific terminology.

Include information about forest types and locations (e.g., country, temperature zone, etc.) in most cases of the cited works.

The information in the last 7 lines on the page 2 should be a part of the Methodology section.

The part of the text starting with "The primary aims of the study..." should be a new paragraph focused exclusively on the aims of the work and/or hypotheses.

Results

For "Aggregation intensity g0-10," provide an exact definition with a citation.

In Fig. 1, the captions for the x-axis (excepting the first column) do not need to be repeated. It is sufficient to show them under the last (lower) axis.

In Fig. 4, follow the same suggestion as in Fig. 1.

In Fig. 5, do the same for the right column of diagrams. Moreover, the legend explaining the colors is unnecessary.

Discussion

The text in the Discussion section needs to be rewritten in proper English, using appropriate terms and adhering to an academic writing style. The main principle for presenting text in the Discussion section is to first present your own results and then discuss findings from other works.

Material and Methods

Provide more details about the forest types included in the study, particularly the tree species composition. While the authors mention the number of species, genera, and families, a table listing the species and their abundances could enhance readers' understanding of the researched forests.

Regarding Fig. 6, it appears that the figures, especially the lower one, depict something different than what the caption states.

Conclusions

The last sentence is vague. The authors should make a stronger statement, perhaps by specifying in more detail the importance of the findings for science, particularly in the field of forest ecology, and if possible, also for potential stakeholders in forestry and/or nature protection.

References

The style of the references is inconsistent (e.g., some papers show the issue number, others do not). Ensure consistency in formatting throughout the references.

Comments on the Quality of English Language

I suggest a major review of the manuscript and the necessity to make detailed linguistic corrections by a native English speaker.

Author Response

Dear reviewer:

Thank you very much for your time involved in reviewing the manuscript and constructive comments on it. We have carefully revised the manuscript by incorporating all the suggestions by the review panel.

Comment 1:

The title: Exploring sustainable development strategies for forests in terms of distribution pattern of living woody and coarse woody debris, warm temperate deciduous broadleaved secondary forests

“It is excessively long and somewhat confusing. Please reconsider it”.

Response 1:

We think this is an excellent suggestion. We have revised the title “Spatial distribution and sustainable development of living woody and coarse woody debris, warm temperate deciduous broadleaved secondary forests in China”.

Comment 2:

In the abstract – “The authors should specify the types of forests, including the species compositions included in the study. Additionally, a strong concluding or generalizing sentence is missing at the end of the Abstract section”.

Response 2:

  • We sincerely thank the reviewer for careful reading. 
  • The 20h sample plot were “warm-temperate deciduous broadleaved secondary forests”. Located on line 2 of the abstract.
  • Species composition, added to the section of Materials and Methods (4.1).
  • Summary sentences were appended to the end of the abstract.

Comment 3:

Introduction

In the first sentence, clarify whether the authors mean tree species specifically or all flora.

"Living woody debris" sounds unusual for traditional scientific terminology.

Include information about forest types and locations (e.g., country, temperature zone, etc.) in most cases of the cited works.

The information in the last 7 lines on the page 2 should be a part of the Methodology section.

The part of the text starting with "The primary aims of the study..." should be a new paragraph focused exclusively on the aims of the work and/or hypotheses.

Response 3:

Thanks for your suggestion.

  • “Species” is a collective term for all species in this study.
  • "Living woody debris" and the removal of the “debris”, means living woody.
  • “Include information about forest types and locations”. The countries and temperature zones were not included in the citation work due to the synthesis of multiple sample plots from the same forest type. These plots may encompass more than one country, and their inclusion could potentially complicate longitudinal comparisons.
  • We greatly appreciate your meticulous examination.
  • "The primary aims of the study..." have been a new paragraph.

Comment 4:

Results

For "Aggregation intensity g0-10," provide an exact definition with a citation.

In Fig. 1, the captions for the x-axis (excepting the first column) do not need to be repeated. It is sufficient to show them under the last (lower) axis.

In Fig. 4, follow the same suggestion as in Fig. 1.

In Fig. 5, do the same for the right column of diagrams. Moreover, the legend explaining the colours is unnecessary.

Response 4:

  • "Aggregation intensity g0-10," the citation, is 29. Another meaning is relative neighbourhood density. Line397-398
  • 1, Fig.4, and Fig.5, coordinates have been modified.

Comment 5:

Discussion

The text in the Discussion section needs to be rewritten in proper English, using appropriate terms and adhering to an academic writing style. The main principle for presenting text in the Discussion section is to first present your own results and then discuss findings from other works.

Response 5:

  • The discussion section has been appropriately restructured.

Comment 6:

Material and Methods

Provide more details about the forest types included in the study, particularly the tree species composition. While the authors mention the number of species, genera, and families, a table listing the species and their abundances could enhance readers' understanding of the researched forests.

Regarding Fig. 6, it appears that the figures, especially the lower one, depict something different from what the caption states.

Response 6:

A description of the composition of tree species has been added to Materials and Methods. Line 359-368

Fig6, changed the title.

Comment 7:

Conclusions

The last sentence is vague. The authors should make a stronger statement, perhaps by specifying in more detail the importance of the findings for science, particularly in the field of forest ecology, and if possible, also for potential stakeholders in forestry and/or nature protection.

Response 7:

Additional additions to the conclusions. Line 461-472

Comment 8:

References

The style of the references is inconsistent (e.g., some papers show the issue number, others do not). Ensure consistency in formatting throughout the references.

Response 8:

The reference layout of the literature was further modified to guarantee uniformity.

Sincerely,

The Authors

Reviewer 2 Report

Comments and Suggestions for Authors

Dear Authors,

I have thoroughly reviewed your paper titled "Exploring Sustainable Development Strategies for Forests in Terms of Distribution Pattern of Living Woody and Coarse Woody Debris in Warm Temperate Deciduous Broadleaved Secondary Forests." The subject of spatial patterns in forest ecosystems, particularly concerning living woody debris (LWD) and coarse woody debris (CWD), is both timely and significant in the context of ecological studies and sustainable forestry practices.

However, I would like to suggest a couple of modifications that I believe could enhance the overall impact and clarity of your paper:

  1. Reorganization of Chapters: Currently, the 'Results' section precedes the 'Materials and Methods' section. I recommend reorganizing these chapters, placing the 'Materials and Methods' before the 'Results'. This restructuring would align with the conventional format of scientific papers, wherein the methodology is explained prior to presenting the findings. It aids in providing the reader with a logical flow, understanding the context and processes of the study before delving into its outcomes.

  2. Detailed Methodological Description: While your study employs advanced statistical models and methodologies, I suggest expanding the description of these methods in greater detail. This elaboration is crucial for two reasons: it enhances the reproducibility of your research, a cornerstone of scientific inquiry, and it provides a clearer understanding for readers who may not be as familiar with these specific models. Detailed methodology will not only increase the transparency of your research but also its educational value for readers who wish to apply similar techniques in their studies.

Implementing these changes, in my view, will not only streamline the narrative of your paper but also bolster its scientific rigor, making it a more valuable resource for researchers in the field of forest ecology and conservation.

Your research is indeed inspiring and contributes meaningfully to our understanding of forest ecosystems. I look forward to seeing these enhancements in your paper, which I believe will further its impact within the scientific community.

Author Response

Dear reviewer:

We appreciate your clear and detailed feedback and hope that the explanation has fully addressed all of your concerns. In the remainder of this letter, we discuss each of your comments individually along with our corresponding responses.

Comment 1:

Reorganization of Chapters: Currently, the 'Results' section precedes the 'Materials and Methods' section. I recommend reorganizing these chapters, placing the 'Materials and Methods' before the 'Results'. This restructuring would align with the conventional format of scientific papers, wherein the methodology is explained prior to presenting the findings. It aids in providing the reader with a logical flow, understanding the context and processes of the study before delving into its outcomes.

Response 1:

  • Your suggestion is commendable as it advocates for placing the technique before the result, hence enhancing the convenience of comprehending the overarching concept of the piece. Materials and Methods are written in accordance with the formatting guidelines of the journal template, after the presentation of the results.

Comment 2:

Detailed Methodological Description: While your study employs advanced statistical models and methodologies, I suggest expanding the description of these methods in greater detail. This elaboration is crucial for two reasons: it enhances the reproducibility of your research, a cornerstone of scientific inquiry, and it provides a clearer understanding for readers who may not be as familiar with these specific models. Detailed methodology will not only increase the transparency of your research but also its educational value for readers who wish to apply similar techniques in their studies.

Response 2:

The proposed idea is very desirable. In the manuscript, the approach was revised by including an explanatory note regarding the model and incorporating the equation.

Line 394-416

Sincerely,

The Authors

Round 2

Reviewer 1 Report

Comments and Suggestions for Authors

The msc has been improved according to my comments.  I have no objections.

Author Response

Dear reviewer:

We are grateful for your thorough review of our paper and the valuable recommendations you have offered. Your expert guidance has been a substantial catalyst and a supportive force for our research endeavours.

We sincerely appreciate the meticulous evaluation you have conducted, and we hold your insightful remarks in high regard.

Sincerely,

MaFang

Reviewer 2 Report

Comments and Suggestions for Authors

Dear authors,

here is recomended structure for papers in MDPI "These are original research manuscripts. The work should report scientifically sound experiments and provide a substantial amount of new information. The article should include the most recent and relevant references in the field. The structure should include an Abstract, Keywords, Introduction, Materials and Methods, Results, Discussion, and Conclusions (optional) sections, with a suggested minimum word count of 4000 words. Please refer to the journal webpages for specific instructions and templates." So my opinion is that you have change order of chapter. It will be better redable

Author Response

Dear reviewer:

Thank you for your detailed and insightful review of our manuscript. We appreciate the time and effort you have dedicated to providing constructive feedback.

We have carefully considered each of your comments and suggestions, and we have made revisions accordingly. Your input has undoubtedly strengthened the quality of our paper. We are grateful for your expertise and guidance.

If there are any additional concerns or aspects you would like us to address, please do not hesitate to let us know. We value your input and look forward to the opportunity to improve our work further based on your feedback.

Thank you once again for your thorough review and valuable contributions to our manuscript.

Sincerely,

MaFang

Round 3

Reviewer 2 Report

Comments and Suggestions for Authors

Thanks. paper is now much better readable.

Author Response

Dear reviewer:

I would like to express my sincere gratitude for your thoughtful and thorough review of my manuscript. Your insights and constructive comments have been invaluable in enhancing the quality and clarity of the paper.

Once again, thank you for your time, expertise, and valuable contributions to the improvement of my manuscript.

Best regards,

MaFang
